# AtomECS: Simulate laser cooling and magneto-optical traps

X. Chen[1], M. Zeuner[2], U. Schneider[2], C. J. Foot[1], T. L. Harte[2], E. Bentine[1*]

**1** Clarendon Laboratory, Department of Physics, University of Oxford, Parks Road, Oxford, OX1 3PU, UK
**2** Cavendish Laboratory, Department of Physics, University of Cambridge, JJ Thomson Avenue, Cambridge, CB3 0HE, UK
* elliot.bentine@physics.ox.ac.uk

May 14, 2021
AION-REPORT/2021-01

## Abstract

**AtomECS is a software package that efficiently simulates the motion of neutral atoms experiencing forces exerted by laser radiation, such as in magneto-optical traps and Zeeman slowers. The program is implemented using the Entity-Component-System pattern, which gives excellent performance, flexibility and scalability to parallel computing resources. The simulation package has been verified by comparison to analytic results, and extensively unit tested.**

# 1 Introduction

Laser cooling uses laser light with a frequency close to resonance with an atomic transition to cool neutral atoms to microkelvin temperatures or lower. Since the earliest demonstrations four decades ago [1–6], magneto-optical traps (MOTs) and Zeeman slowers have become established work-horses that are widely used to produce cold atoms. Laser cooling has enabled numerous advances, from precision measurements of time [7] and acceleration [8, 9] to the

quantum simulation of novel phases of matter [10–12].

The flux of atoms from the initial laser-cooling stage is an important parameter in the design of an ultracold atom experiment, as a low flux can limit the rate at which experiments run. When designing an apparatus, simulations are an important tool for optimising the performance and characteristics of Zeeman slowers and magneto-optical traps [13–20]. The performance of such techniques depends on many parameters, and the fraction of incident atoms that are captured and cooled may be less than $0.1\,\%$, so computational speed is critical when using simulations to optimise and explore the wide parameter space. The calculation time can be greatly reduced by using simplified models of the laser cooling process.

AtomECS simulates the motion of atoms interacting with near-resonant laser light, as is used for laser cooling and in magneto-optical traps. The atomic trajectories are calculated by integrating the effect of the scattering forces exerted on atoms in a configuration of laser beams, using a rate-equation approach. AtomECS is written in Rust and follows the Entity-Component-System architectural pattern, which gives excellent performance and extensibility [21–23]. The accuracy of the program is verified by a suite of unit tests and comparisons to simple physical scenarios with analytic results.

This paper is structured as follows. We begin in Section 2 with a review of the main features of laser cooling, stating which of these are presently included in AtomECS. Section 3 describes the multi-beam rate-equation method AtomECS uses to calculate scattering forces. Section 4 presents a high-level overview of the AtomECS package, followed by extensions in Section 5 and worked examples in Section 6 that illustrate a range of physical scenarios. Section 7 examines the implementation of AtomECS in Rust, while Section 8 quantifies the parallel performance of the program.

## 2 Physics of laser cooling

Photons transmit both energy and momentum. When an atom absorbs a photon it receives a momentum kick of $\hbar\mathbf{k}$, where $\mathbf{k}$ is the wavevector of the incident photon. The excited atom then decays back to the ground state by the spontaneous emission of another photon in a random direction. For appropriately chosen conditions the spontaneously emitted photon carries away more energy than the atom gained in the stimulated absorption process and thus, averaged over many such events, the kinetic energy of the atom is dissipated. This principle underlies methods that cool atoms through the scattering of photons, such as laser cooling in Zeeman slowers and magneto-optical traps.

In the following sections we discuss some notable features of laser cooling and their level of support in AtomECS; for a more detailed treatment of the physics we refer the reader to [24, 25].

### 2.1 The photon scattering rate

The rate $R_i$ at which an atom in the ground state undergoes stimulated absorption of photons from a laser beam $i$ is,

$$R_i = \frac{I_i(\vec{x})}{I_{\text{sat}}} \frac{\Gamma}{2} \frac{\Gamma^2/4}{\delta(\vec{x}, \vec{v})^2 + \Gamma^2/4},$$ (1)

where $I_i$ is the intensity of the laser beam at the atom's position $\vec{x}$, $\Gamma$ is the linewidth, related to the lifetime of the excited state $\tau$ by $\Gamma = 1/\tau$ for a two-level system, $\delta = \omega_L - \omega_0$ is the angular frequency detuning between the photon ($\omega_L$) and the atomic transition ($\omega_0$), and $I_{\text{sat}}$ is the saturation intensity, defined as the intensity for which light with $\delta = 0$ gives $1/2$ of the absorption of a low intensity beam of light. $R_i$ is maximum at resonance, for which $\delta = 0$.

## 2.2   Velocity-dependent damping forces

The angular frequency detuning $\delta(\vec{x}, \vec{v})$ depends on atomic velocity because of the Doppler effect, which causes a moving atom to perceive a shift in the frequency of the laser light. An atom with a component of velocity in the direction opposite to that of a laser beam experiences a positive shift in the frequency of the light. Thus, an atom in a pair of counter-propagating red-detuned laser beams preferentially scatters photons from the beam that opposes its motion, resulting in a velocity-dependent force that slows the motion of the atom. A frequently used configuration consists of three orthogonal pairs of red-detuned counter-propagating beams, which provides damping forces along all directions of atomic motion. This is known as optical molasses [24]. The Doppler shift is implemented in AtomECS.

## 2.3   The magneto-optical trap (MOT) and position-dependent forces

In the presence of an inhomogeneous magnetic field, $\delta(\vec{x}, \vec{v})$ also depends on position because the magnetic field causes a Zeeman shift of the optical transition. Adding a weak magnetic field gradient to the optical molasses configuration causes an imbalance in the scattering rates from each counter-propagating beam when an atom is displaced from the magnetic field node. This produces a position-dependent restoring force that confines the atoms, and is known as a magneto-optical trap (MOT). The magnetic field gradients are too weak to produce magnetic confinement by themselves. The Zeeman shift is implemented in AtomECS. Although AtomECS describes each atomic transition as a two-level system, Zeeman shifts are calculated explicitly for each laser beam.

## 2.4   Loss of atoms from the cycling transition

For many species of atoms the electronic transitions used for laser cooling do not form a completely closed cycle; an excited atom may relax into a state outside of the cycling transition that does not scatter photons from the cooling beams and is therefore no longer laser cooled. A common solution is to employ additional *repump* light at a frequency that excites these atoms, which may then relax back into a state involved in the cycling transition. In many cases, the forces resulting from the comparatively few scattered repump photons are sufficiently small to be neglected. AtomECS supports the loss of atoms out of the laser-cooling transition.

## 2.5   Rescattering of photons, forces between atoms

In dense clouds of cold atomic vapour there is strong absorption of light, such that photons that are spontaneously emitted by some atoms may be re-absorbed by other atoms before exiting the cloud. A photon scattered by one atom and rescattered by a second atom produces net impulses on each that are equal and opposite. The rate of rescattering between a pair of atoms scales as $1/r^2$, where $r$ is their separation, resulting in a repulsive $1/r^2$ long-ranged force [26, 27]. Long-ranged attractive forces also result from the formation of shadows in the

incident beams due to scatter from optically-thick atom clouds, which reduces the intensity of light downstream. Neither rescattering nor shadows are currently implemented in AtomECS. These long-ranged forces were recently considered in detail by Gaudesius *et. al.* [19].

## 2.6  The Doppler limit

Although the average force applied to an atom by counter-propagating beams in the MOT tends to zero as the velocity decreases, its variance does not. This is due to fluctuations in the number of photons absorbed from each beam. Furthermore, each photon absorption is followed by emission in a random direction, which increases the variance of atomic velocities. In the absence of sub-Doppler cooling mechanisms, these fluctuations impose a lower bound on the temperatures that are achievable with Doppler cooling [6, 28],

$$T_D(\delta) = \frac{\hbar\Gamma}{2k_B} \frac{1 + (2\delta/\Gamma)^2}{4\,|\delta|\,/\Gamma}, \tag{2}$$

where $T_D(\delta)$ is the Doppler temperature, $\delta$ is the angular detuning of the beams from resonance, and Eq. (2) holds in the low intensity limit. $T_D(\delta)$ has a minimum value at $\delta = -\Gamma/2$, for which $T_D = \hbar\Gamma/2k_B$. AtomECS intrinsically respects the Doppler limit.

## 2.7  The recoil limit

The laser cooling process requires the spontaneous emission of at least one photon, which imparts a recoil impulse of $\hbar k$ to the atom in a random direction. These fluctuations lead to the recoil temperature limit,

$$T_{\mathrm{r}} = \frac{\hbar^2 k^2}{2m k_B}, \tag{3}$$

where $m$ is the mass of an atom. For many species $T_{\mathrm{r}}$ is much smaller than the Doppler limit, but the recoil limit is important for MOTs operating on narrow-linewidth transitions for which $T_D < T_{\mathrm{r}}$ [29], such as the 689 nm transition in strontium [30,31]. Laser cooling below the recoil limit has also been demonstrated [32, 33] but these methods are not implemented in AtomECS.

AtomECS simulates the recoil from individual photons, similarly to Ref. 17, and so captures some aspects of recoil-limited behaviour. However, the rate-equation method used in AtomECS is not adequate for modelling the behaviour of multilevel atoms undergoing laser cooling near the centre of a magneto-optical trap. More sophisticated calculations using optical Bloch equations (OBEs) can capture some of the features, but they are semi-classical calculations which cannot describe all the details, for instance when atoms have momentum comparable to that of a single photon of the incident radiation. Comprehensive treatments have been implemented using fully quantum calculations that also quantise the momentum [34].

## 2.8  Sub-Doppler cooling

Various methods exist to cool atoms below the Doppler limit. These rely for instance on the polarisation gradients that arise from the overlap of counter-propagating beams with crossed polarisations, and optical pumping between ground-state sublevels on a timescale longer than

the radiative lifetime of the atom [35]. These sub-Doppler cooling mechanisms are currently not supported by AtomECS.

## 2.9   Multi-level systems and molecules

AtomECS does not consider the complex level structures of molecules and multi-level systems. To model such systems see Refs. 13, 14.

# 3   The rate equation method

AtomECS uses the rate equation method to calculate the forces experienced by an atom that scatters photons from a given configuration of laser beams. Each laser beam $i$ has wavevector $\mathbf{k}_i$ and intensity $I_i$. The atom is modelled as a classical, coupled two-level system, with energy levels separated by $\hbar\omega_0$. The population densities $\rho_{gg}$ and $\rho_{ee}$ for the ground and excited states obey

$$
\begin{aligned}
\dot{\rho}_{ee} &= -\rho_{ee}A_{21} - \left(\sum_i R_i\right)(\rho_{ee} - \rho_{gg}), \\
\dot{\rho}_{gg} &= +\rho_{ee}A_{21} + \left(\sum_i R_i\right)(\rho_{ee} - \rho_{gg}).
\end{aligned}
\tag{4}
$$

These equations incorporate the effects of both stimulated ($R_i$) and spontaneous ($A_{21}$) processes. $A_{21}$ is the Einstein A coefficient for decay from the upper to lower level, and is equal to the linewidth $\Gamma$ for a two-level system. For a two-level system, $I_{\mathrm{sat}} = \hbar\omega_0^3\Gamma/12\pi c^2$. We include the Zeeman shifts and rate for $\sigma_{\pm}$- and $\pi$-transitions when calculating the total stimulated rate for each beam. The polarisation of the light relative to the direction of the local magnetic field is treated as in Ref. 27. The rate equations are solved in the limit $t \to \infty$ to give steady-state solutions of the population densities:

$$
\begin{aligned}
\rho_{ee} &= \frac{\sum_i R_i}{A_{21} + 2\sum_i R_i}, \\
\rho_{gg} &= 1 - \rho_{ee}.
\end{aligned}
\tag{5}
$$

Stimulated absorption followed by emission into the same beam imparts no total force, so we restrict our consideration to processes that consist of stimulated absorption followed by spontaneous emission, and neglect coherences between the beams. The average total number of photons absorbed and then spontaneously emitted by the atom over a duration $\Delta t$ is equal to

$$
N_\gamma = \Delta t \rho_{ee}\Gamma,
\tag{6}
$$

which reaches a maximum value of $\Gamma\Delta t/2$ when the two-level system is saturated.

To determine the scattering force applied during each timestep, we take the expected

number of photons scattered by each laser beam $N_i$ as

$$N_i = \frac{R_i}{\sum_i R_i} N_\gamma, \tag{7}$$

where following from Ref. 15 we have distributed the total scattered photons between the beams, weighted by $R_i$. The accuracy of this multi-beam rate equation approach is discussed in Appendix A. The expected average $N_i$ are converted into the actual numbers of photons scattered during the timestep, $\tilde{N}_i$, by drawing from Poisson distributions with means $N_i$. Subsequently, the forces applied to the atom by the stimulated absorption and spontaneous emission of $\tilde{N}_i$ photons from each beam are calculated. The forces arising from stimulated absorption act in the direction of the laser field,

$$F_{\text{SA},i} = \frac{\hbar \mathbf{k}_i \tilde{N}_i}{\Delta t}. \tag{8}$$

We assume that the spontaneous emission is isotropic[1] and causes a random walk in momentum space,

$$F_{\text{SE},i} = \sum_j^{\tilde{N}_i} \frac{\hbar |\mathbf{k}_i|}{\Delta t} \mathbf{e}_j, \tag{9}$$

where $\mathbf{e}_j$ is a unit vector that points in a random direction. When the number of photons scattered is large, the net force due to $\tilde{N}_i$ individual photon emissions can be approximated by applying a single force drawn from a Gaussian distribution with standard deviation $\sqrt{\tilde{N}_i/3}(\hbar |\mathbf{k}_i| /\Delta t)$.

# 4 Overview of AtomECS

AtomECS provides a set of features that can be used to simulate laser cooling of neutral atoms. In this section we provide a breakdown of the core features of AtomECS. Throughout this paper, rust code examples will be indicated by `this font`.

## 4.1 Optical scattering forces

AtomECS supports optical scattering forces arising from absorption and emission, from an arbitrary number of laser beams with Gaussian intensity profiles. Forces are calculated using the rate equation model of Section 3, and so include the effects of saturation and power broadening, but do not include the effects of interference between coherent beams. The calculated interaction between atoms and light includes Zeeman shifts and Doppler shifts, which are essential for simulating a magneto-optical trap. Fluctuations in the number of

---

[1]Spontaneous emission is not isotropic and its angular distribution depends on the polarisation of the radiation exciting the atomic dipole. This detail has a minor effect that could be taken into account for configurations with a single laser beam such as a Zeeman slower, however with more complex 2D and 3D light field configurations the assumption that the average over the polarisation of all laser beams gives approximately isotropic emission is a good approach.

photons scattered during each timestep and random forces caused by recoil from photon emission can be enabled (see Section 5.2), giving rise to the Doppler limit.

## 4.2 Magnetic fields

AtomECS supports calculations of magnetic fields using analytic formulae for 2D and 3D quadrupole fields, and for uniform bias fields. For more complicated field geometries, Atom-ECS supports the use of magnetic field values defined explicitly over a regular 3D grid.

## 4.3 Atom sources

Atoms can be added to an AtomECS simulation using a number of approaches:

1. Placement of single atoms into the simulation domain. This approach is used for simple examples, and for unit tests so that the initial state of each atom is well defined.

2. Creation of atoms using an oven. Atoms are spawned at the aperture of the oven, with a $v^3$-Boltzmann velocity distribution that depends on the oven temperature. The angular distribution follows the $j(\theta)$ function of Ref. 36, which is derived from a free molecular flow model of atoms travelling through microchannels in the oven aperture.

3. Emission of atoms from the surface of a simulation volume, as if desorbed from the surface of a vacuum chamber. The velocities follow a Maxwell-Boltzmann distribution determined by the temperature of the surface, and the angular distribution assumes a Lambert cosine distribution based on the interior normal of the surface.

4. Gaussian sources, which generate atoms whose positions and velocities follow user-defined Gaussian distributions.

# 5 Extensions

The AtomECS package is modular by design, and many features can be enabled or disabled depending on simulation preference. For instance, to determine the capture velocity in a MOT it may be useful to disable random fluctuations (and thus the Doppler limit) so that the atoms only experience mean forces to reduce computation time. The various configuration options are described below.

## 5.1 Velocity limit

The velocities of atoms emitted from ovens or present in thermal vapor cover a wide range of values. Many of these are fast-moving atoms (e.g. $v > 100\,\mathrm{m\,s^{-1}}$), which cannot be captured by laser cooling. To prevent wasting CPU effort simulating these trajectories, an optional velocity limit can be defined by adding a `VelocityCap` to the simulation. An atom is discarded upon creation if its velocity exceeds the cap value. This optimisation greatly decreases run time with no loss of accuracy if an estimate of the capture velocity of an experiment is known.

## 5.2  Fluctuations in the scattering force

Random fluctuations in the number of photons scattered by atoms each frame give rise to heating effects as discussed above. These fluctuations are configured to be enabled (default) or disabled by adding a `ScatteringFluctuationsOption` resource to the AtomECS simulation, as shown in the example of Section 6.3.

Fluctuations also arise from the random momentum kicks imparted when an atom relaxes from an excited state and emits a photon. These emission forces are configured by adding an `EmissionForceOption` to the simulation, and are enabled by default.

## 5.3  Loss of atoms from the cycling transition

To include the loss of atoms out of the laser cooling cycle, add a `BranchingRatio` component to the atoms. Each time a photon is absorbed or emitted by an atom in the simulation, the `BranchingRatio` is used to determine whether that atom relaxes into a state that does not interact with the cooling light. Such atoms are tagged with a `Dark` component and ignored from future interactions with the cooling beams.

## 5.4  Simulation volumes

Simulation volumes are used to discard atoms that stray out of bounds. They are composed from geometric shapes, such as `shape::Cuboid`s and `shape::Cylinder`s, and are defined as either inclusive or exclusive by a `sim_region::VolumeType`. Each integration step, atoms are deleted if they are either: not within any inclusive region; or are inside an exclusive region. Using simulation volumes optimises the simulation by removing atoms that have escaped the region of interest, e.g. those that have left the MOT capture region.

## 5.5  Configuring output

A `FileOutputSystem` periodically writes atomic components such as position, id or velocity to an output file, in either a binary or text format. Some examples of configuring file output are given below. The `ConsoleOutputSystem` prints the timestep and atom number to the console every 100 timesteps.

# 6  Worked examples

In this section we present a selection of the example simulations that accompany the AtomECS package, and gradually introduce the features available in AtomECS. Where possible, results are compared to analytic forms to demonstrate the accuracy of generated data.

AtomECS follows the Entity-Component-System (ECS) data-oriented programming pattern, implemented through the Rust crate `specs`. The three ingredients of an ECS pattern are as follows: an *Entity* identifies a 'thing' in the simulation, for example an atom; *Components* are containers for data that are associated with individual entities and represent aspects of those entities, for example position; *Systems* operate on components and entities to implement behaviour by transforming the component data, or by adding/removing components to/from entities. In AtomECS, systems are responsible for performing integration, calculating

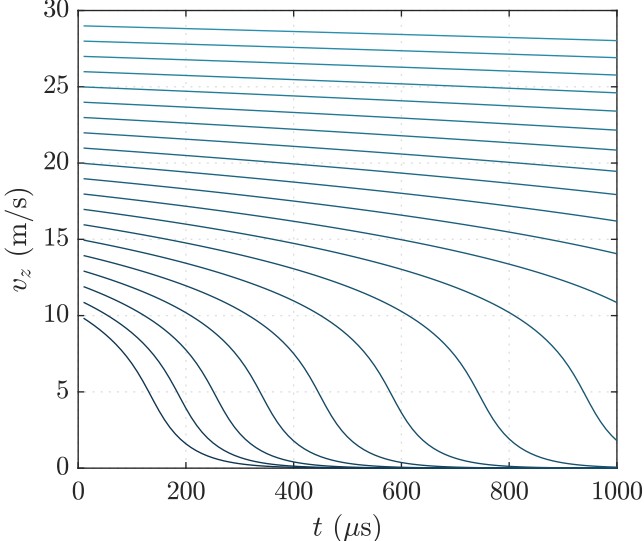

Figure 1: Velocity of $^{87}$Rb atoms in a one dimensional optical molasses. The beams have a power of $10\,\text{mW}$ each, a $1/e$-radius of $1\,\text{cm}$ and a detuning of $-12\,\text{MHz}$. Each line corresponds to a different initial velocity.

fields and forces, creating atoms, and generating file output. Further details can be found in Section 7.1.

## 6.1 One-dimensional optical molasses

The first example simulates $^{87}$Rb atoms in a one-dimensional optical molasses, formed by two counter-propagating laser beams that are red-detuned from the D2 cooling transition. For this first example, we will *not* include fluctuations in the number of scattered photons, nor the random kicks arising from the recoil from spontaneously emitted photons. Figure 1 shows example trajectories generated by this program, demonstrating that the velocities are reduced over time. The simulation source can be found in `optical_molasses_1d.rs`.

We first create the dispatcher that will schedule the systems that update the simulation world and perform the numerical integration. This dispatcher is created by a builder object:

```
let mut builder = ecs::create_simulation_dispatcher_builder();
```

Additional systems are added using the builder's `add` command. We add a system that outputs velocities to a text file every 10 frames.

```
builder.add(
  file::new::<Velocity, Text>("vel.txt".to_string(), 10),
  "",
  &[],
);
```

Finally, we invoke the `build()` method to create the dispatcher object, then use it to add any required resources (such as component storages) to the simulation world.

```
let mut dispatcher = builder.build();
dispatcher.setup(&mut world.res);
```

We now populate the simulation world with entities that represent the physical scenario, starting with the atoms of rubidium. Each atom consists of an entity to which the required 'atom like' components are attached:

```
world
  .create_entity()
  .with(Position {
    pos: Vector3::new(0.0, 0.0, -0.03),
  })
  .with(Atom)
  .with(Force::new())
  .with(Velocity {
    vel: Vector3::new(0.0, 0.0, 10.0 + (i as f64) * 5.0),
  })
  .with(NewlyCreated)
  .with(AtomicTransition::rubidium())
  .with(Mass { value: 87.0 })
  .build();
```

The `AtomicTransition` component holds properties of the laser cooling transition, here configured to be the $^{87}$Rb 780 nm D2 line, and we set the `Mass` to 87 u.

Similarly, the cooling lasers are added by creating entities and attaching the 'laser like' components.

```
world
  .create_entity()
  .with(GaussianBeam {
    intersection: Vector3::new(0.0, 0.0, 0.0),
    e_radius: 0.01,
    power: 0.01,
    direction: -Vector3::z(),
  })
  .with(CoolingLight::for_species(AtomInfo::rubidium(), -12.0, -1.0))
  .build();
```

Here, the `GaussianBeam` component describes the spatial profile of the cooling beam, which points along the negative $z$ direction, intersecting the origin, and has $1/e$-radius of 0.01 m and total power of 10 mW. The `CoolingLight` component describes the frequency and polarisation of the light, which here is 12 MHz red-detuned from the rubidium cooling transition and with a $\sigma_-$ polarisation.

The timestep duration is configured by adding a `Timestep` resource to the world.

```
world.add_resource(Timestep { delta: 1.0e-6 });
```

In this example, we use $\Delta t = 1\,\mu s$.

The simulation is performed through an integration loop. This loop alternately invokes the dispatcher (to calculate forces and integrate motion) and `world.maintain()` (to clean the world by adding or deleting new entities or modifying component composition).

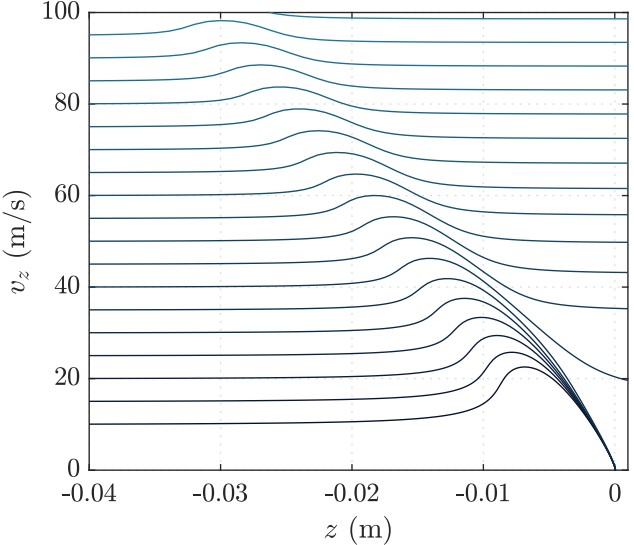

Figure 2: Plot of velocity $v_z$ versus position $z$ in the 1D MOT. Each line corresponds to a different initial velocity. The capture velocity is around $35\,\mathrm{m\,s^{-1}}$.

```
for _i in 0..5000 {
    dispatcher.dispatch(&mut world.res);
    world.maintain();
}
```

## 6.2   One-dimensional MOT

The 1D molasses is transformed into a 1D MOT by the addition of a magnetic field gradient. We use a quadrupole field of the form $\vec{B} = B'(x, y, -2z)$, with $B' = 15\,\mathrm{G\,cm^{-1}}$. The field is added as an entity with both `QuadrupoleField3D` and `Position` components, where the `Position` component encodes the location of the quadrupole node (here, at the origin).

```
world
    .create_entity()
    .with(QuadrupoleField3D::gauss_per_cm(15.0, Vector3::z()))
    .with(Position::new())
    .build();
```

Atoms are placed into the simulation at $z = -4\,\mathrm{cm}$, with velocities in the range $10\,\mathrm{m\,s^{-1}}$ to $110\,\mathrm{m\,s^{-1}}$. Figure 2 shows example trajectories in the phase-space $(z, v_z)$. Atoms travelling slower than the capture velocity (here $35\,\mathrm{m\,s^{-1}}$) are slowed and trapped by the MOT.

## 6.3   Doppler temperature in a 3D MOT

Next, we simulate the motion of atoms in a six-beam MOT, using the example `doppler_limit.rs`. The configuration consists of three perpendicular pairs of counter-propagating beams as is

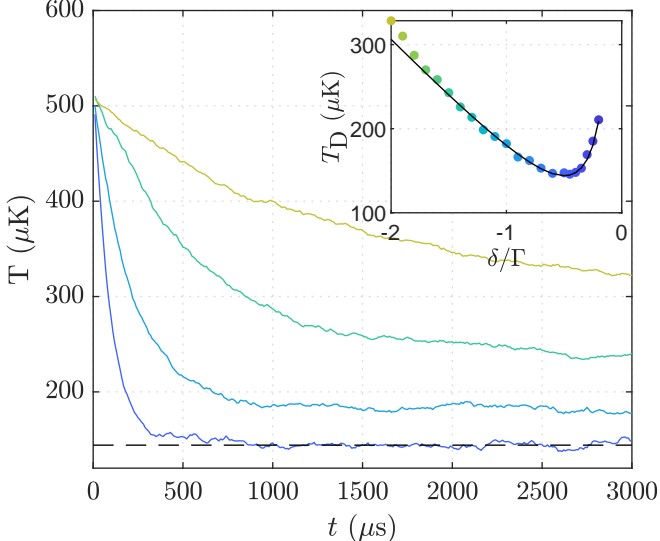

Figure 3: Temperature of $^{87}$Rb atoms captured in a three-dimensional magneto-optical trap. Different series correspond to different values of detuning, indicated by the color shown in the inset, and the minimum Doppler temperature of $144\,\mu\text{K}$ at $\delta = -\Gamma/2$ is indicated by the dashed line. Inset: Temperature of simulated rubidium MOT versus detuning (points), shown with an analytic result for the Doppler temperature (line).

common for magneto-optical trapping, with beams aligned to the Cartesian axes. The magnetic field consists of a quadrupole field with cylindrical symmetry about the $\hat{\mathbf{e}}_z$ axis.

We enable fluctuations by adding two resources to the simulation.

```
world.add_resource(
        EmissionForceOption::On(
                EmissionForceConfiguration {
                        explicit_threshold: 5,
                }
        ));
world.add_resource(ScatteringFluctuationsOption::On);
```

The first option enables random kicks caused by the recoil from the spontaneous emission of photons. These kicks are applied individually when the number of recoils each timestep is below a threshold (here, 5). For performance reasons, when the number of recoils is above the threshold we instead represent the total recoil by a single impulse sampled from a Gaussian distribution, see Section 3. The second option enables fluctuations in the number of photons scattered by an atom during each frame; the number of photons to scatter is drawn from a Poisson distribution with mean determined by the rate equation approach, see Eq. (7).

Figure 3 shows the temperature of simulated ensembles as a function of time, relaxing towards the Doppler limit. At small time scales, the temperature undergoes small fluctuations due to random variations in the number of scattered photons, and the random forces arising from spontaneous emission. The Doppler limit depends on the detuning of the cooling light [6], and in the inset of Figure 3 we show good agreement of simulation temperatures with a

theoretical calculation of the Doppler limit over a range of detunings.

## 6.4   Optimisation of a 2D-MOT source

This section explores the use of AtomECS to optimise a source of laser-cooled atoms. The source uses a 2D MOT to cool $^{88}$Sr atoms emitted from an oven, and a *push* beam to direct the cooled atoms towards a second region. There are a multitude of packages which can be used for the optimisation process itself; we use the Matlab `bayesopt` function to perform a Bayesian optimisation. Both Matlab and Rust code for this example can be found on a separate repository at `https://github.com/TeamAtomECS/` `source_optimisation_example`.

The optimisation loop proceeds as follows. First, Matlab writes a serialised file containing the simulation parameters to run. We use the `json` format because it is both widely used and supported. Five parameters are optimised, consisting of: the push beam power, radius, and detuning; the quadrupole gradient; and the detuning of the cooling light. Next, Matlab invokes the `cargo` command line program to run the simulation. AtomECS deserialises the input file and integrates the equations of motion to generate atomic trajectories. Finally, the generated data are analysed by Matlab to extract a measure of performance (the *cost function*), which is used to steer the optimiser towards better parameters.

The optimisation loop attempts to minimise the cost function, which we set equal to the negative of the fraction of atoms that are captured and cooled by the 2D-MOT source. Motivated by a typical experiment geometry, we define an atom as *captured* if it reaches a 2 cm wide region located 20 cm from the centre of the 2D-MOT source, and perpendicular to the direction of the oven. Minimising the cost function therefore corresponds to optimising the flux of laser-cooled atoms.

Each simulation begins with 1 million atoms ejected from the oven. Any atoms travelling faster than the velocity cap of $230\,\mathrm{m\,s^{-1}}$ are immediately removed, which leaves typically 100,000 atoms remaining. Their motion is integrated for 30 ms of motion. Each simulation takes around 4 s to complete on a standard desktop using a 6-core i7-8700 CPU.

Figure 4a shows the results of running this optimisation for 3 hours. The observed peak performance improves over time as the optimiser explores the parameter space. Figure 4b plots the trajectories of cooled atoms in position space for the final optimised parameters.

## 7   Implementation details

### 7.1   Entity-Component-System

ECS is an architectural pattern which follows the principle of behaviour by composition. When a system operates on the simulation world, it acts upon entities which possess particular combinations of components. For example, let us consider an `UpdatePositionSystem` which moves entities according to their velocities. This system operates on entities which have both a `Position` and `Velocity` component, reading from `Velocity` and read/writing to `Position`. If the `Velocity` component were removed from an entity, it would no longer be affected by this system. Mutating the composition of an entity allows complex behaviours to be produced; a common pattern is to use some component types as tags which signal that systems should process particular entities, e.g. a `Fixed` component could be used to indicate an entity should be ignored by our `UpdatePositionSystem`. This approach differs

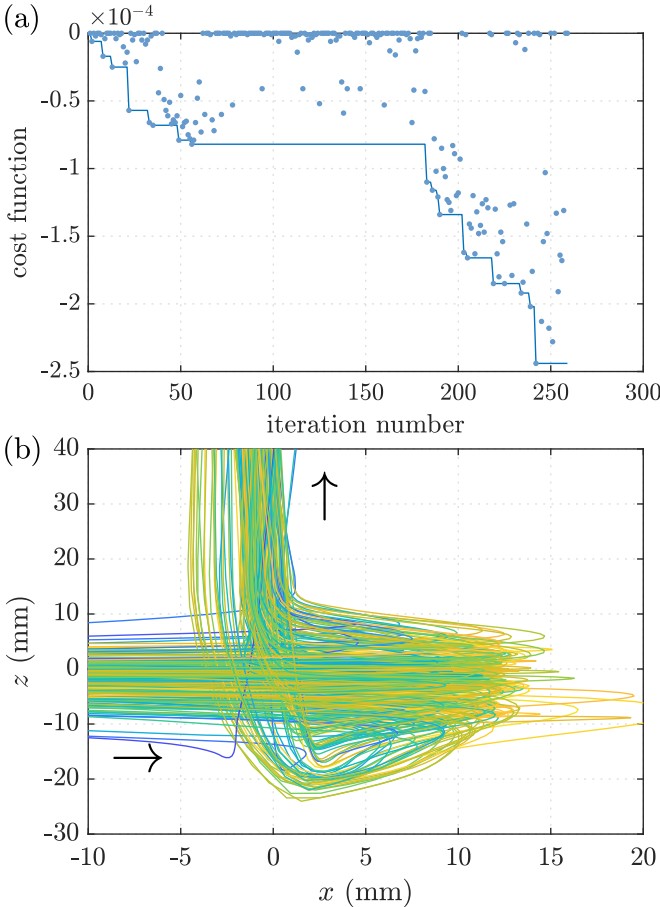

Figure 4: Optimisation of a 2D-MOT source using AtomECS. (a) Evolution of the cost function after successive iterations. The line indicates the observed minimum while dots show the results of individual simulations. (b) Trajectories of captured atoms for the final optimised parameters, in the vicinity of the 2D MOT. Atoms enter from the left (oven 8 cm away), are cooled by the MOT beams, then ejected by a push beam in the $+z$ direction. Only the captured atoms are shown, corresponding to 221 of the original 1 million simulated atoms. Each trajectory is color-coded by the initial velocity, ranging from $0\,\mathrm{m\,s^{-1}}$ (blue) to $140\,\mathrm{m\,s^{-1}}$ (yellow).

fundamentally from behaviour by inheritance models that are typically used in some Object-Oriented Programming.

A great strength of the ECS pattern is that it produces flexible, loosely-coupled code - the systems communicate only via component data, and individual systems can be enabled or disabled to add or remove functionality to the simulation. Another advantage is that programs written in this style can be easily extended to parallel execution; systems are explicit about the data they require read/write access to, so dependency and parallelism become easy to resolve. Finally, the ECS pattern yields a program memory structure that is well-suited to high-performance computing. Common components are stored in contiguous arrays, so a system requesting components can quickly shuffle the relevant memory into and out of the processor cache. This memory structure contrasts to the randomly allocated 'heaps' more typical of managed-memory applications.

In recent years, the ECS pattern has become increasingly popular in the video games industry, where both flexibility and performance of software are primary concerns [21–23]; for instance Unity, one of the largest game engines by market share, has begun migrating time-critical functionality to the ECS pattern [21].

## 7.2   Structure of an AtomECS simulation

Having described the principles of ECS, we now move to the implementation details in Atom-ECS.

The vast majority of entities in a typical AtomECS simulation are the atoms whose trajectories we seek to calculate (examples of non-atom entities include magnetic coils and atom sources). Table 1 outlines some of the components associated with atom entities. These components hold the information required to perform force calculations and to integrate each atom's trajectory. Some components are used as flags to filter the entities that systems should operate on. For instance, the `NewlyCreated` component indicates an atom has been created within the current integration step, and so instructs relevant systems to initialise properties of the new atom.

## 7.3   The simulation loop and the dispatcher

During each timestep in the simulation loop, the AtomECS systems perform calculations to modify data stored in the atom components and integrate the equations of motion, as detailed in the following sections. The dispatcher manages and orders the execution of these systems to achieve system-level outer parallelism; when two systems do not require conflicting access to the same component storages, i.e. they write to different components, then those systems are automatically executed in parallel (see Section 8.1). The duration of a timestep is set by the `Timestep` resource.

After the systems have run, `world.maintain()` is invoked to process structural changes to the entities, such as those that occur when new atoms are generated.

## 7.4   Step initialisation

Each step in the loop begins by initialising the values of components which aggregate quantities, e.g. the `ClearForcesSystem` sets the value of all `Force` components to zero. Subsequent systems will add to the `Force` components, to calculate the total force applied to each

atom during the timestep.

## 7.5    The magnetic systems

The systems in the `magnetic` module calculate the magnetic field at each atom's `Position` and store the result in the atom's `MagneticFieldSampler` component, so that it may be used for later calculations. Table 2 describes the role of each system in this module.

## 7.6    The laser systems

Systems in the `laser` module calculate the optical scattering forces applied to each atom. Table 3 gives a description of the systems used to implement the rate equation method of Section 3.

## 7.7    Velocity-Verlet integration

The atomic trajectories are integrated using the velocity-Verlet algorithm, which updates the positions and velocities as

$$x_{i+1} = x_i + v_i \delta t + \frac{a_i}{2} \delta t^2,$$
$$v_{i+1} = v_i + \frac{a_i + a_{i+1}}{2} \delta t, \tag{10}$$

where $x_i, v_i$ and $a_i$ are the position, velocity and acceleration of an atom at timestep $i$. This first-order method requires only one force calculation each step. The velocity-Verlet minimises integration error when forces depend only on position. Although this is not true of the velocity-dependent forces in laser cooling, the effect of integration error manifests as a small heating term which is negligible compared to the rate of cooling. We have chosen velocity-Verlet to minimise the integration error for cases with forces that depend only on atom positions, such as future simulations of evaporative cooling of atom clouds in conservative external potentials (dipole force and magnetic traps). These cases lack an external cooling mechanism to mitigate the build up of integration error.

## 7.8    Atom generation

New atoms are added to the simulation by systems in the `atom_sources` module. Out of the box, AtomECS supports atomic sources such as ovens, or thermal atoms spawned on the surfaces of simulation regions.

AtomECS represents a source of atoms (e.g. an oven) as an entity. The `Oven` component determines the temperature, direction of emission and aperture properties. Other components associated with source entities include the source `Position`, the number of atoms to emit, and properties of the generated atoms such as mass distributions and details of the laser cooling transition. Sources can emit atoms continuously during the simulation, or emit once by adding a `ToBeDestroyed` component to the source to delete it after the first step.

Table 4 outlines the systems used to generate atoms. These systems insert atoms into the world using the `LazyUpdate` resource, and therefore requires a call to `world.maintain` to complete the process. All new atoms are created with `Force`, `Mass` and `Atom` components,

and additionally marked with a `NewlyCreated` component for one time step, which allows other modules to detect new atoms and add further components to them.

## 7.9 Tests and verification

The AtomECS code is extensively unit tested to guarantee that each individual system is correctly implemented and consistent with the laws of physics. Integration tests verify that the full simulation program correctly calculates forces when the individual systems are combined. All tests are run from the `cargo` command line tool using `cargo test --release`.

# 8 Performance

## 8.1 Parallelism

Parallelism is implemented in two ways within AtomECS. The first is system-level parallelism, whereby systems that require different component resources can operate concurrently. A specific example of this within AtomECS are `SampleLaserIntensitySystem` and `CalculateDopplerShiftSystem`; the first writes to `LaserIntensitySamplers` components, while the second writes to `DopplerShiftSamplers`. As these two systems do not require simultaneous write access to the same storages they can execute concurrently on separate threads. This concurrency is automatically handled by the dispatcher for all systems. The second type of parallelism is inner-system parallelism, where loops within systems are iterated in parallel by a pool of worker threads. The parallel loops are implemented using specs `par_join()`, which in turn uses the `rayon` Rust crate.

## 8.2 Benchmarking

The benchmark simulation calculates the trajectories of $N_a$ atoms after $N_s = 5000$ steps, with all simulation features included. The simulation uses a thread pool of size $N_t$ and takes a total of $T_{\text{sim}}$ seconds to complete. We define the *normalised time* $\tau = T_{\text{sim}}/N_a N_s$, which is the average wall time per timestep per atom, as a measure of performance scaling. An idealised, perfectly parallel program would have $\tau$ independent of $N_s$, $N_a$ and inversely proportional to $N_t$, but any real implementation suffers an overhead associated with scheduling the thread pool.

Figure 5 shows the calculated value of $\tau$ for a range of different thread and atom numbers, running on a machine with a 6-core i7-8700 CPU. For small numbers of atoms $\tau$ is higher because there is negligible gain from parallel operations compared to simulation overheads. As the number of threads increases, $\tau$ decreases in accordance with Amdahl's law [37], with best-fit giving a proportion $1-p=15\%$ of the program which is not parallelised.

# 9 Conclusion

AtomECS is a high-performance, flexible program for simulating the laser cooling of neutral atomic vapors, and is appropriate for the simulation of cold atom sources. Furthermore,

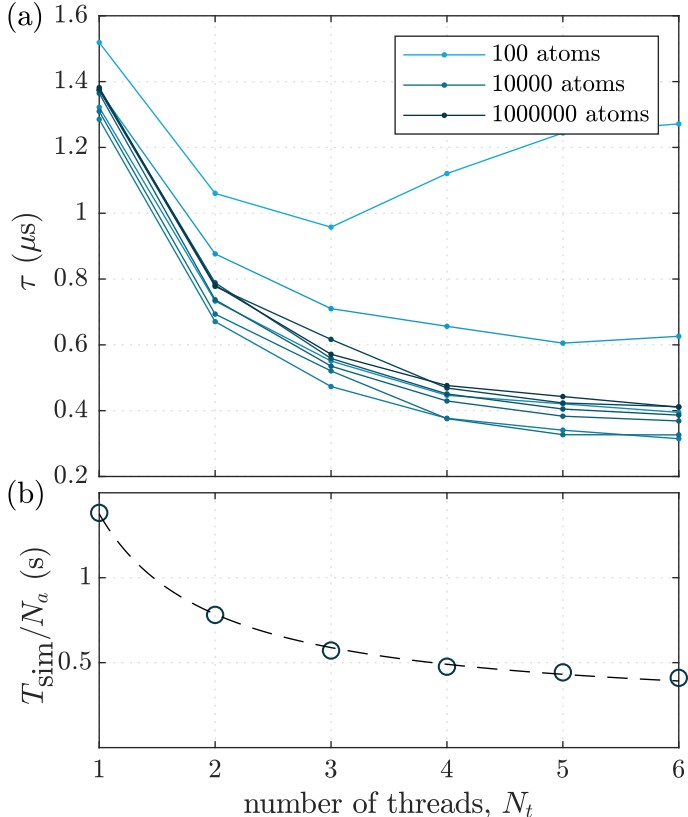

Figure 5: (a) Normalised time $\tau$ as a function of thread and atom number. The number of atoms is indicated by the color of the line, with 9 values logarithmically spaced and ranging from 100 atoms (blue) to 1,000,000 atoms (black). (b) Total wall time per step as a function of number of threads for 1,000,000 atoms. The best-fit line for Amdahl's law with $p = 0.15$ is shown by the dashed line.

AtomECS has a number of configurable features to enable the user to balance simulations between performance and physical accuracy.

We have explained the physical model used for calculating scattering forces and described their implementation details within AtomECS. A number of program examples have been given to demonstrate how to use AtomECS and to showcase the flexibility - these range from simple physical examples to a full optimisation of an apparatus.

AtomECS is implemented using the Entity-Component-System architecture, giving it great flexibility and performance. Unit tests provide an automated means to test that new feature modifications do not change or break existing functionality. The results of simulations have also been compared to theoretical predictions. AtomECS is open source, and we welcome contributions from other users. The code is under active development: work is currently underway to implement conservative dipole-force potentials; s-wave collisions; and long-ranged forces that arise from rescattering photons in the MOT.

## 10    Getting Started

AtomECS is available at `https://github.com/TeamAtomECS/AtomECS`, along with general instructions in the accompanying `readme`. The Rust command line tools are required to build AtomECS, and can be found on the Rust website.

## 11    Acknowledgements

This work was supported by the EPSRC, Grant References EP/S013105/1 and EP/P009565/1. EB, TLH, MZ, US, CJF acknowledge the Atom Interferometer Observatory and Network (AION) project. The authors acknowledge useful discussions about the ECS pattern with Tim Watts and Robin Rexstedt, and about laser cooling with Mike Tarbutt.

## A    Comparing the rate equation and optical Bloch equation approaches

In this section we test the validity of the multi-beam rate equation approach to calculate the scattering forces, which was described in Section 3. We compare the scattering forces to those arising from an optical Bloch equation (OBE) treatment, and also to the forces from an independent-beam rate equation model, in which saturation parameters and forces are calculated for each beam without considering the effect on saturation from the intensities of the other laser beams in the radiation field..

We consider an atom moving in the fields of a transverse-loaded two-dimensional MOT source, as demonstrated for lithium [38], sodium [39] and strontium [40] atoms. The optical fields are comprised of circularly-polarised fields co- and counter-propagating along the axes

$\hat{\mathbf{e}}_x$ and $\hat{\mathbf{e}}_y$,

$$E(x, y, z) = A e^{i(\omega t - kx)} \begin{pmatrix} 0 \\ 1 \\ i \end{pmatrix} + A e^{i(\omega t + kx)} \begin{pmatrix} 0 \\ 1 \\ -i \end{pmatrix}$$
$$+ A e^{i(\omega t - ky)} \begin{pmatrix} 1 \\ 0 \\ i \end{pmatrix} + A e^{i(\omega t + ky)} \begin{pmatrix} 1 \\ 0 \\ -i \end{pmatrix}.$$

Near the origin, the magnetic field is $B(x, y) = B'(x\hat{\mathbf{e}}_x - y\hat{\mathbf{e}}_y)$, producing a Zeeman splitting $\hbar\omega_B = g_J \mu_B B$. Atoms enter the MOT region from an oven, oriented in the direction $(\hat{\mathbf{e}}_x + \hat{\mathbf{e}}_y)/\sqrt{2}$.

The optical transition from $|J = 0\rangle \to |J' = 1\rangle$, has one ground $(m_J = 0)$ and three excited sublevels $(m_{J'} = 0, \pm 1)$, as in the case of $\mathrm{Sr}^{88}$ laser-cooled on the $461\,\mathrm{nm}$ transition. The Hamiltonian for this two-level, four-state system is the sum of Hermitian matrices representing a free evolution term,

$$\mathcal{H}_0 = \hbar \begin{pmatrix} 0 & & & \\ & \omega_0 - \omega_B & & \\ & & \omega_0 & \\ & & & \omega_0 + \omega_B \end{pmatrix},$$

and an interaction term,

$$\mathcal{H}_{\mathrm{int}} = \frac{\hbar}{2} \begin{pmatrix} 0 & \Omega_- & \Omega_\pi & \Omega_+ \\ \Omega_-^\dagger & 0 & & \\ \Omega_\pi^\dagger & & 0 & \\ \Omega_+^\dagger & & & 0 \end{pmatrix}.$$

The time-dependent couplings $\Omega(t)$ are determined by transforming the electric fields into a frame aligned to the local B-field at the position of the atom, and projecting the field into $\sigma_\pm$- and $\pi$-polarised components.

The time evolution of the density matrix is given by

$$i\hbar \frac{\partial \rho(t)}{\partial t} = [\mathcal{H}, \rho(t)]. \tag{11}$$

Spontaneous emission causes the excited-state populations $\rho_{22}, \rho_{33}, \rho_{44}$ to decay into the ground state $\rho_{11}$ at a rate $\rho_{ii}\Gamma$, while off-diagonal coherences decay as $\Gamma/2$ [41]. Including these decays into Eq. (11) gives the Optical Bloch Equations (OBE) [24]. Solutions to the OBE exhibit transients that decay over a timescale $1/\Gamma$.

To model the total flux of atomic sources, we are interested in the behaviour of faster atoms at the edge of the laser beams, which determines the capture velocity of the source and hence the fraction of incoming thermal atoms that are laser cooled. These atoms behave akin to a two-level system, because the large Doppler and Zeeman shifts $(kv, \omega_B \gg \Gamma)$ make one transition near resonance and the others detuned. Near the origin the magnetic field is weak and the Zeeman shift is small $(\omega_B \ll \Gamma)$, so slowly-moving atoms $(kv \ll \Gamma)$ interact with all polarisation components of light. The resulting population dynamics involve all three

sub-levels of the excited $J = 1$ state.

Let us now consider the scattering forces that contribute to the laser-cooling process. These arise from stimulated absorption followed by spontaneous emission, and occur at a rate $\Gamma \rho_{ii}$ for an excited state $i$. In Figure 6 these forces are calculated using three different models: the optical Bloch equations; the multi-beam rate equation model of Section 3; and an independent-beam rate equation model that considers saturation from each beam independently.

In the top panel of Figure 6 we calculate the force due to a single circularly-polarised beam. For this scenario the rate equations are exact solution to the OBEs, hence there is exact agreement between these models. The independent- and multi-beam rate equation methods are also equivalent, because there is only one beam. In the high intensity limit $(I \gg I_{\text{sat}})$, the excited state population converges to $\rho_{ii} = 1/2$, producing a force $\hbar k \Gamma / 2$.

In the bottom panel of Figure 6b, we consider an atom with position $x = y, z = 0$, propagating with velocity $\vec{v} = v(\hat{\mathbf{e}}_x + \hat{\mathbf{e}}_y)$ in the optical fields of the 2D MOT. The two beams counter-propagating along $-\hat{\mathbf{e}}_x$ and $-\hat{\mathbf{e}}_y$ have equal Doppler shift and both interact with the atom. The average force due to balanced absorption from these beams, followed by spontaneous emission, is equal to $F = -\hbar k \Gamma \rho_{44}(\hat{\mathbf{e}}_x + \hat{\mathbf{e}}_y)/2$. In the high intensity limit, the multi-beam rate model and OBE model have good agreement, but the independent-beam rate model over-estimates the force because it assumes the atom scatters light at a limiting rate of $\Gamma/2$ photons/s for each beam. For low intensities $(I \ll I_{\text{sat}})$ there is good agreement between all models. Magneto-optical traps of neutral atoms typically use beams where $I \sim I_{\text{sat}}$, for which the multi-beam rate model offers the best approximation to the OBE model.

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

| Component | Description |
|---|---|
| `Atom` | Tag, marks an entity as an atom. |
| `Mass` | The mass of the atom, in atomic mass units. |
| `Position` | Position of the entity, SI units of m. |
| `Velocity` | Velocity of the entity, SI units of m/s. |
| `Force` | Force on the entity, SI units of N. |
| `AtomicTransition` | Characterises the cooling transition, e.g. frequency and linewidth. |
| `NewlyCreated` | Tag that indicates the entity has just been created. |
| `ToBeDestroyed` | Tag that indicates the entity should be removed from the simulation. |
| `Dark` | Tag that indicates that an atom does not interact with cooling light. |
| `MagneticSampler` | Stores information about the magnetic field at the location of this entity; populated by systems in the `magnetic` module. |
| `LaserDetuningSampler`, `DopplerShiftSampler`, `ZeemanShiftSampler` | Stores information about the interaction of an atom with optical fields, such as the Doppler shift for each beam. Data populated by systems in the `laser` module. |
| `RateCoefficients` | Rate coefficients $R_i$ for all beams as defined in Eq. (1) |
| `TwoLevelPopulation` | Represents the steady-state population density of the excited state and ground state |
| `TotalPhotonsScattered` | Holds the total number of photons that the atom is expected to scatter in the current simulation step from all beams. |
| `ExpectedPhotons` `ScatteredVector` | Array of mean expected numbers of photons scattered by the atom for each beam |
| `ActualPhotons` `ScatteredVector` | Array of actual numbers of photons scattered by the atom for each beam, during this timestep |

Table 1: An overview of components attached to atoms during a typical AtomECS simulation.

| System | Description |
|---|---|
| `ClearMagneticFieldSamplerSystem` | Sets the value of all `MagneticFieldSampler` components to zero. |
| `Sample3DQuadrupoleFieldSystem` | For each entity with `Position` + `MagneticFieldSampler`, calculates the magnetic field from entities with `Position` + `QuadrupoleField3D`. Writes the result to the sampler. |
| `UniformMagneticFieldSystem` | For each entity with `MagneticFieldSampler`, calculates the magnetic field from entities with `UniformMagneticField`. Writes the result to the sampler. |
| `SampleMagneticGridSystem` | For each entity with `Position` + `MagneticFieldSampler`, calculates the magnetic field from entities with `PrecalculatedMagneticFieldGrid`. Writes the result to the sampler. |
| `CalculateMagneticFieldMagnitudeSystem` | For each entity with a `MagneticFieldSampler`, calculates the magnitude of the field vector and stores it in the sampler. |
| `AttachFieldSamplersToNewlyCreatedAtomsSystem` | Schedules `MagneticFieldSampler` components to be added to `NewlyCreated` atoms at the end of the simulation step. |

Table 2: Overview of systems and components used by the `magnetics` module to calculate magnetic fields. The systems are listed in order of execution priority, but systems may be executed in parallel when they do not have conflicting read/write to the same component stores. The boxes indicate `read only` and `read/write` components.

| System | Description |
|---|---|
| `DopplerShiftSystem` | Calculates the doppler shift with respect to each beam due to atom `Velocity` and stores the result in the `DopplerShiftSampler`. |
| `ZeemanShiftSystem` | Calculates the Zeeman shift of the transition for each atom using the `MagneticFieldSampler` and stores the result in the `ZeemanShiftSampler`. |
| `GaussianIntensity System` | Samples the intensity of each beam at each atom's `Position` and stores the result in the `LaserIntensitySampler`. |
| `DetuningSystem` | Uses the `ZeemanShiftSampler` and `DopplerShiftSampler` to calculate the angular detuning of each beam, stores the result in `LaserDetuningSampler`. |
| `RateCoefficientSystem` | Calculates $R_i$ using `LaserDetuningSampler` and `LaserIntensitySampler`, stores the result in `RateCoefficients`. |
| `TwoLevelPopulation System` | Determines the steady-state population densities $\rho_g$ and $\rho_e$ using `RateCoefficients` and Eq. (5). The population densities are stored in `TwoLevelPopulation`. |
| `MeanTotalPhotonsSystem` | Calculates the expected number of photons scattered $N_\gamma$ by using `TwoLevelPopulation` and Eq. (6). The result is stored in `TotalPhotonsScattered`. |
| `MeanPhotonsSystem` | Calculates the expected number of photons scattered by each beam $N_i$, by using `TotalPhotonsScattered`, `RateCoefficients` and Eq. (7). The result is stored in `ExpectedPhotonsScattered`. |
| `NumberPhotonsSystem` | Calculates the actual number of photons scattered by each beam $\tilde{N}_i$, by using `ExpectedPhotonsScattered` and drawing from a Poisson distribution. The result is stored in `ActualPhotonsScattered`. |
| `AbsorptionForceSystem` | Uses the `ActualPhotonsScattered` to apply a force due to absorption $\tilde{N}_i$ photons from each beam during the timestep. The result is stored in `Force`. |
| `EmissionForceSystem` | Uses the `ActualPhotonsScattered` to apply a force due to isotropic emission of $\tilde{N}_i$ photons from each beam during the timestep. The result is stored in `Force`. |

Table 3: An overview of `systems` and `components` used for the AtomECS rate equation implementation. The boxes indicate `read` and `read/write` component access. The systems are listed here in priority of execution, although some may be executed in parallel when they do not have conflicting read/write access.     29

| System | Description |
|---|---|
| `EmitNumberPerFrame System` | Calculates the `AtomNumberToEmit` this frame using `EmitNumberPerFrame`. Used for continuous sources. |
| `EmitFixedRateSystem` | Calculates the `AtomNumberToEmit` for sources with a fixed rate of emission using `EmitFixedRate` and the `TimeStep` resource. When the ratio of rate to timestep duration is not an integer, random numbers are drawn that give the correct average rate. |
| `PrecalculateForSpecies System` | Pre-calculates constant quantities (such as probability distributions) of a `MaxwellBoltzmannSource` and `MassDistribution` to accelerate atom generation. The result is stored in a `PrecalculatedSpeciesInformation` component added to the source entity. This system processes a source once. |
| `PrecalculateForGaussian SourceSystem` | Pre-calculates constant quantities of a `GaussianVelocity-DistributionSourceDefinition` and stores the result in the `GaussianVelocityDistributionSource`. |
| `OvenCreateAtomsSystem` | Generates `AtomNumberToEmit` atoms from an `Oven` using the `AtomicTransition`, `Position` and `PrecalculatedSpeciesInformation` components. |
| `CreateAtomsOnSurface System` | Generates `AtomNumberToEmit` atoms on the surface of a `SurfaceSource`'s `<Volume>`, using the `AtomicTransition`, `Position` and `PrecalculatedSpeciesInformation` components. |
| `GaussianCreateAtoms System` | Generates `AtomNumberToEmit` atoms with given `Mass` and `AtomicTransition` according to a `GaussianVelocityDistributionSource`. |
| `EmitOnceSystem` | Sets `AtomNumberToEmit` to zero for sources with `EmitOnce` component. |
| `CentralCreatorCreate AtomsSystem` | Creates atom entities from a `CentralCreator` source with custom direction, velocity, mass and position distributions. |

Table 4: An overview of `systems` and `components` used for atom sources in AtomECS. The boxes indicate `read` and `read/write` component access. The systems are listed here in priority of execution, although some may be executed in parallel when they do not have conflicting read/write access.