# Peer review of "AtomECS: Simulate laser cooling and magneto-optical traps"

_SciPost Physics Codebases_

## Round 1 · Referee Report · Anonymous (Referee 1) · 2021-10-10

Strengths

1- introduced to the community an example of a trendy programming architecture (entity-component-system) for high performance simulation. 2- the high performance would enable a simulation-based (pre-)optimization of system parameters of a complex cold-atom experiment, which could have immediate impact for existing and future setups. 3- possibility to simulate a magneto-optical trap from the atomic source such as an oven. 4- the benchmark of the parallel performance is demonstrated.

Weaknesses

1- complex level structure (beyond two-level) is not yet supported.

Report

The manuscript "AtomECS: Simulate laser cooling and magneto-optical traps" reports a new software package for numerical simulation of atomic trajectories under laser cooling. While many implementations exist for various purposes, it is computationally difficult to use such simulations to optimize system parameters of an experiment. This package employs a modern architecture of data-oriented programming that allows highly efficient simulations of large systems. Certain simplifications are employed such as limiting to two-level system and the rate-equation method. Nevertheless, it can well simulate a MOT consistent with the Doppler limit, even directly from an atomic beam source. It is then possible to simulate a complete setup (from oven to MOT chamber) and use it to optimize its design for example. The manuscript is well structured, generally well written and well referenced. It meets the criteria of the journal and I recommend its publication in SciPost Physics Codebases. I do suggest more details to be added on the project repository and a few changes that may improve the readability.

Specific comments: 1- It is not yet clear to me if the rate-equation method is sufficient for the narrow-line MOT. Could the authors comment on this? 2- Is it possible to implement the OBE solution as an option? In this case, would it significantly reduce the efficiency or there are other issues? 3- Given that there are existing packages for similar tasks, it would be very helpful if the authors could provide a short "selection guide" summarizing some pros and cons/limitations of a few packages (known to the authors) such that one could more easily find the optimal solution. 4- Instructions about the installation (including relevant information about rust) are scarce (on the github page). The authors are encouraged to supplement these instructions for a broader user base. 5- In 6.1, the code and the explanation only mentions a beam pointing along -z. This is a bit confusing as another beam pointing along z also has to be defined. 6- In 6.4, I think it would be helpful to include the code that shows how the atom source is defined. Additionally, it would be interesting to see how an atomic beam defined two orifices could be coded. 7- In appendix A at the very end, $v$ used for the comparison should be specified (although from the context it is in the regime $kv\gg\Gamma$). It would also be interesting though to see comparison in the other regime.

Requested changes

suggested changes see comments in the report

---

## Round 1 · Referee Report · Anonymous (Referee 2) · 2021-12-14

Strengths

Very clear introduction, introducing basic concepts of laser cooling and trapping based on radiation pressure and its statistical properties, and linking them to the code.
Very clear distinction between what the code can do and what it can't.
Opening possibility to explore experimentally relevant situations by simulations, with quantitattive estimates.

Weaknesses

The application to real systems is very limited. No interference, no polarization effects, no multilevel atoms. Hence, many physically interesting cases are not covered, or, maximally, estimates can be derived.

Report

I recommend publication, because the code is certainly of pedagogical value and may also turn out useful for quantitative estimates of experimental situations, like in the examples provided.

Requested changes

Some perspective should be given what future extensions will provide, and which limitations may not be overcome, because of the simplified physical ansatz.

---

## Editorial Decision

awaiting_resubmission